# Mechanistic Concept of Physiological, Biochemical, and Molecular Responses of the Potato Crop to Heat and Drought Stress

**DOI:** 10.3390/plants11212857

**Published:** 2022-10-26

**Authors:** Milan Kumar Lal, Rahul Kumar Tiwari, Awadhesh Kumar, Abhijit Dey, Ravinder Kumar, Dharmendra Kumar, Arvind Jaiswal, Sushil Sudhakar Changan, Pinky Raigond, Som Dutt, Satish Kumar Luthra, Sayanti Mandal, Madan Pal Singh, Vijay Paul, Brajesh Singh

**Affiliations:** 1ICAR-Central Potato Research Institute, Shimla 171001, India; 2ICAR-Indian Agricultural Research Institute, New Delhi 110012, India; 3ICAR-National Rice Research Institute, Cuttack 753006, India; 4Department of Life Sciences, Presidency University, 86/1 College Street, Kolkata 700073, India; 5ICAR-Central Potato Research Institute Campus, Jalandhar 144026, India; 6ICAR-Central Potato Research Institute Campus, Modipuram 250110, India; 7Department of Biotechnology, D. Y. Patil Arts, Commerce and Science College, Sant Tukaram Nagar, Pimpri, Pune 411018, India

**Keywords:** *Solanum tuberosum*, heat tolerance, drought tolerance, bulking, climate change

## Abstract

Most cultivated potatoes are tetraploid, and the tuber is the main economic part that is consumed due to its calorific and nutritional values. Recent trends in climate change led to the frequent occurrence of heat and drought stress in major potato-growing regions worldwide. The optimum temperature for tuber production is 15–20 °C. High-temperature and water-deficient conditions during the growing season result in several morphological, physiological, biochemical, and molecular alterations. The morphological changes under stress conditions may affect the process of stolon formation, tuberization, and bulking, ultimately affecting the tuber yield. This condition also affects the physiological responses, including an imbalance in the allocation of photoassimilates, respiration, water use efficiency, transpiration, carbon partitioning, and the source–sink relationship. The biochemical responses under stress conditions involve maintaining ionic homeostasis, synthesizing heat shock proteins, achieving osmolyte balance, and generating reactive oxygen species, ultimately affecting various biochemical pathways. Different networks that include both gene regulation and transcription factors are involved at the molecular level due to the combination of hot and water-deficient conditions. This article attempts to present an integrative content of physio-biochemical and molecular responses under the combined effects of heat and drought, prominent factors in climate change. Taking into account all of these aspects and responses, there is an immediate need for comprehensive screening of germplasm and the application of appropriate approaches and tactics to produce potato cultivars that perform well under drought and in heat-affected areas.

## 1. Introduction

Potato (*Solanum tuberosum* spp. *tuberosum* L.) is the third most important food crop after rice and wheat, with increasing popularity in terms of human consumption. Its annual production was 388.19 million tons (MT) in the year 2019, which is expected to increase further in several regions [1]. Potato is a popular staple vegetable in many countries. Potato is also a staple food in European countries, adding carbohydrates to the human diet and nutrients and minerals [2]. As the human population continues to rise, the accessibility of food may emerge as a major concern on a worldwide scale, and thus potato can help to provide food and nutritional security [3]. This crop is also vital in light of ongoing climate change, which is already exerting intense pressure on the human population’s food and grain supply. In the coming decades, climate change will become a major rising problem for governments and policymakers to devise ways to combat the adverse effects of climate change and ensure food and nutritional security [4].

About 10,000 years ago, the domestication of potato took place in the highlands of the Andes in South America [5]. In the early 16th century, explorers from European countries, such as Spain, England, and the Netherlands, introduced potatoes to Europe [6]. Potatoes are grown extensively in two primary regions. The first zone is between 45° N and 57° N, where potato is cultivated as a summer crop; while the second zone is the subtropical lowlands between the latitudes of 23° N and 34° N, where potato is cultivated during the winter [7]. In the subtropical and tropical regions, the potato is grown as a winter crop where the night temperatures remain below 22 °C. However, a temperature below 20 °C is usually required for tuberization in potato [8]. The term “tropicalization” refers to the process of breeding and developing suitable production techniques for vegetable crops that may be grown at lower latitudes [9].

The importance of potatoes in securing food and nutritional security was identified by the Food and Agriculture Organization (FAO) of the United Nations when it declared the year 2008 as the “The International Year of the Potato” [10]. This initiative was pursued to attract the world’s attention toward the importance of potatoes and their more significant role in food and nutritional security in nonconventional areas. Most developing countries are on the Asian and African continents, where the production and demand for potatoes have increased in recent years [11]. The tropicalization of the potato crop is vital, as potato serves as a cheap source of energy and nutrition. The potato can help overcome food and nutritional insecurity and contribute to improving economic growth [12].

Environmental factors such as heat, drought, salinity, flood, and cold are the major causes of adverse effects on the growth, development, and productivity of horticultural crops [13]. Abiotic stressors are the major cause of crop loss on a global scale, since they can reduce the average yield of most crops by more than 50 percent [14]. The rise in global temperature is a threat to agriculture in general. The increase in temperature poses significant abiotic stress for crop plants that adversely affects their survival, adjustment, and performance [12]. Under such a changing climate scenario, potato cultivars need to be developed that can thrive under high temperatures and give reasonably good production and productivity [15].

Heat stress negatively affects the plant’s growth, and developmental, biochemical, and physiological processes, leading to a reduction in yield and productivity. The critical developmental stage affected by heat stress is the reproductive and bulking stages [16]. Similarly, plant response to drought stress is detrimental and impacts morpho-physiological, anatomical, and biochemical parameters. Climate change is detrimental to tuber and root vegetable crops, which are also considered staple foods in many countries. It has been predicted that there will be a decrease in global potato production by 18–32% due to global warming by the middle of this century [12,17]. Therefore, to cope with this climate change problem and ensure food security, it is essential to understand the responses of crops to climate change. Understanding the mechanism of plant response to abiotic stresses could be a viable strategy for developing crop varieties through selection, breeding, and biotechnological approaches, with the goal of developing varieties tolerant to heat or/and drought stress [18]. Regions in tropical areas suffer from unprecedented seasonal heat and drought stress [19,20]. These stresses lead to a detrimental effect on physiological and biochemical mechanisms in the plant that ultimately hamper the growth and development of potato plants [21] and reduce yields and tuber quality [22]. The potato crop originated in borderline subtropical/alpine climates and performed best in places with warm days and cool nights. Potato production in tropical and sub-tropical regions is challenging. However, potato breeders and physiologists are trying to develop thermo-insensitive varieties which may be suitable for tropical regions [23].

Potato cultivation is expanding to non-traditional regions with water-deficient conditions and facing heat stress. In addition, heat and drought spells are becoming more frequent in temperate zones [12,16]. Thus, an integrated approach is required to understand better the morphological, physiological, biochemical, and molecular network at the whole-plant level pertaining to drought and heat stress tolerance. This is then used to develop potato cultivars suitable for abiotic stress conditions. This review highlights various physiological, biochemical, and molecular aspects and responses of potato plants under tropical conditions where the potato plant is exposed to several types of abiotic stresses.

## 2. Production and Productivity of Potato Affected by Heat and Drought Stress

Potato crop species are highly prone to different abiotic (high-temperature stress, drought, salinity, and mineral stress) and biotic stresses (insect and pest attacks) [13,24,25,26]. Heat stress is a significant issue for temperate countries and potato production locations in the semi-arid Middle East and the Sub-Saharan, subtropical, and tropical regions [27]. Temperature is the most critical uncontrollable factor affecting potato growth, development, production, and productivity. Tropical areas experience high-temperature stress, where plants undergo several anatomical, morphological, physiological, biochemical, and molecular changes. Growth and development are seriously affected, leading to a substantial decrease in potato production [16]. Due to high temperatures, the environment of the tropical region alters the morphological features and the physiological and developmental processes of potato plants. For instance, high temperatures may cause a reduction in the leaf area index, specific leaf area, size and number of leaves, and canopy development, an increase in the plant’s lateral branching and height, and a decrease in the number and size of tubers. The alarming rate of increase in temperature due to climate change causes more frequent heat stress to plants during the summer and high night temperatures in the winter, which hampers crop yield and quality in any region of the world [27]. Heat stress mediates imbalances in source–sink activity, allocation of photoassimilates, necrosis, and malformation of tubers [27]. Furthermore, the soil temperature in which the potato is grown affects the process of stolon formation, tuberization, and bulking, ultimately reducing the tuber yield [28,29].

High-temperature and water stress conditions affect the yield and quality of potato tubers, where the severity, duration, and timing of both heat and water stress adversely affect sprout emergence, stolon formation, tuberization, and final yield of the potato tubers [30]. The potato tuber yield depends on tuber bulking [29], which occurs in the late stage of growth. The production may decrease due to bulking reduction, which is affected by heat and water-deficient conditions [28]. As per Obiero et al. [30], the high-temperature treatment affects the whole plant’s dry matter and potato tuber yield. They reported that high-temperature treatment (30 °C) compared to a control (temperature of 22 °C) before and after tuber initiation leads to 45% smaller tubers (less than 2.5 cm diameter).

Planting time (spring and autumn) affects potato yield [31]. High temperatures in the subtropical climate, particularly during the spring season, are more detrimental than autumn because of the combination of low humidity (higher atmospheric evaporative demand) and higher temperatures. An average yield reduction of 68% and 42% was observed in spring and autumn plantings [22]. This difference may also be due to water availability during the spring and autumn seasons [32]. Under tropical conditions, high-temperature stress negatively impacts potato tuber quality and yield through the inhibition of the transport of photoassimilates to the developing stolons [12]. It was reported by Fleisher et al. [33] that the optimum temperature for photosynthesis and biomass accumulation in potato is 20 °C. Additionally, it was reported that the optimum daily mean temperature might be as low as 13 °C [12]. An increase of every 5 °C above the optimum temperature causes a reduction in the photosynthetic rate by 25%, which ultimately affects biomass accumulation and, later, the sink activity [15,30,34].

As previously documented, potato production decreases due to heat and drought stress in most East African countries [35]. Additionally, Jarvis et al. [36] anticipated about a 15% reduction in potato yield in Africa by 2030. Across Asia, India and China are at constant drought and heat stress risk. Moreover, periods of high temperatures and drought are becoming more frequent in Central and Western Europe. US potato production was also severely affected due to drought and heat stress during the last 2–3 years. In Mediterranean regions, the problem of dry spells in potato cultivation is also a major concern [37]. Likewise, the major potato-growing areas of the world under consistent risk of drought and heat stress are highlighted in Figure 1.

## 3. Physiological Changes and Responses of Potato Plants under Heat and Drought Conditions

The changing climate in temperate, subtropical, and tropical regions affects the physiological process of the potato plant [38]. As potato is now adopted for the tropical region, it faces high-temperature and drought stress. Thus, potato plants grown under these conditions may have to acclimate to the situation and alter their growth and development process; however, the final response depends mainly on the intensity of high-temperature and/or drought stress [39]. The adverse effects of the combination of high-temperature and drought conditions lead to a reduction in canopy mass of the potato plant, reduction in photosynthesis and water use efficiency, chlorophyll degradation (Table 1), and an acceleration of leaf senescence [40]. The average optimum daily temperature conditions for potato stolon formation and tuberization are 15–20 °C [8,27,41]. Temperatures above 22 °C have an adverse effect on vegetative, as well as on the reproductive, growth of the potato plant [33,42]. The tolerance to high-temperature stress in potato depends mainly on some critical parameters such as the genotype, developmental stage at which it is exposed, level of stress faced, and the ability to form stolons, initiate tubers, and bulk [15,30,34]. Other physiological aspects such as anatomical features (morphology of vascular bundles), total biomass production, respiration, transpiration, carbon partitioning, and the source–sink relationship are dependent on extrinsic, as well as intrinsic, factors of the potato plant [43,44,45]. Besides physiological responses, biochemical responses are also presented in Table 1.

### 3.1. Anatomy and Morphology

The major challenge for the breeder is to make the potato suitable for the tropical climate with maintained production [27]. In this direction, when selecting and developing suitable potato cultivars for these areas, they need to be phenotyped for tolerance/resistance to various abiotic stresses [54]. The tropicalization of potatoes is mainly affected by heat and drought stresses, which are known to alter the anatomical structure and morphology of the potato plant [55,56]. The development of vascular bundles under high-temperature stress may affect the transport of nutrients and photoassimilates in the xylem and phloem, respectively [34]. It was noticed that the cultivars with a reduction and enhancement in the size of the xylem and phloem, respectively, can better withstand high-temperature conditions. The increased size of the phloem enhances the sink capacity to store more photoassimilates and convert them into starch, which ultimately increases tuber yield [34]. In the tropical climate, high temperatures lead to the development of tall plant with small leaves, thin stems, long stolons, increased internodes (elongated internodes), inhibition of tuber development, and reduction in the ratio of tuber fresh weight to total fresh weight [57].

The leaf is the primary source of potato plants that might be affected in many ways under various abiotic stresses [42]. Under heat stress (30 °C), the leaf area of the potato plant was reported to reduce by 35% compared to the control (22 °C). Several wild cultivars of potato are known to be cultivated extensively in altitudes between 2000 to 4000 m (above sea level), where the season is generally represented by long day length, high light intensity, and cooler temperatures. However, the globalization of potato was achieved by different voyagers in different parts of the world. The modern cultivars were selected over time and adopted for long-day conditions. The branching pattern of the shoot in the potato plant was also reported to be affected under high-temperature stress [30]. The growth of lateral shoot branches was affected by heat stress. The number of lateral branches and their diameter were reported to be enhanced under high-temperature stress. The canopy of the potato plant also changes under high temperatures. As the temperature rises over 23 °C, the number of axillary branches increases, resulting in more leaves and a faster rate of senescence. Recent reports suggested that the length of the main branch was reduced under heat stress, which ultimately leads to the stunted growth of the potato plant (Table 1). The change in the ratio of growth hormones such as auxin and gibberellins is responsible for developing lateral and main shoots [58]. Moreover, potato genotypes with the stay-green trait may better alleviate the adverse effects of heat and drought stresses [59].

The root is the important organ responsible for water and mineral nutrient uptake and is highly influenced by abiotic stress [60]. The average optimal temperature required for the growth of roots, tubers, and stolons varies with the growth stages of the plant [8]. The root is the first organ that senses water-deficient conditions and responds accordingly based on physiological, biochemical, and molecular phenomena. The root system architecture and its morphology in potato are significantly affected by both heat and drought stress [61]. The modification of the root architecture occurs by forming more lateral roots and root hairs [62]. High-temperature stress (33 °C) also delays adventitious root initiation in potato. Water-deficient conditions affect potato root system morphology by increasing the proliferation of lateral roots, root thickness, inhibition of root elongation, and increasing root hair formation [63]. Potato cultivars with a deep root architecture may prove beneficial in combating both drought and high-temperature stress.

### 3.2. Role of Photoperiod

The photoperiod substantially affects the potato’s developmental stages, from the emergence through tuber initiation phases. Short photoperiods (10–12 h) enhance the tuber initiation compared to longer photoperiods (14–18 h). Moreover, it has been previously documented that the photoperiod has little or no effect after tuber initiation. Thereafter, temperature plays a crucial role in the tuberization process [31]. Higher temperatures are inhibitory to tuberization, irrespective of photoperiod, viz., short-day and long-day conditions, although the adverse effect is much more significant in long photoperiods. Higher temperatures affect the partitioning of the assimilates by reducing the amount delivered to tubers and enhancing the content of other parts. These effects are observed in both long and short photoperiods. The longer photoperiods coupled with heat and drought stress adversities might significantly delay and reduce tuberization along with excessive vegetative growth of the haulm [31,33]. Thus, potato crops face a higher yield reduction in heat and drought stress-affected areas encountering shorter day lengths (mainly subtropical lowlands).

### 3.3. Carbon Partitioning and Source–Sink Relationship

Potato is an indeterminate crop with respect to its growth habit, where the vegetative growth can continue even after flowering and tuber formation; this reduces the sink ability and activity [33]. Drought and heat stress also affect the source–sink relation, carbon partitioning, and potato tuber development [64]. Environmental stress often impacts these parameters, which affects the photosynthetic rate, xylem and phloem transport, sugar metabolism, and photoassimilate diversion away from sink tissues [43,64,65]. Any stress during the early stages is particularly damaging because it decreases carbon assimilation, reduces partitioning, and ultimately impacts tuberization, bulking, and final tuber production [28]. The relationship between the source and sink can be disturbed by high-temperature stress, which further delays the process of tuberization and can result in tuber necrosis and deformities. When the temperature rises above the optimum, the photoassimilates are translocated away from the tuber [66].

As potato originated in the hills of the Andes, it has the inherent ability to tuberize in temperature ranges between 14 °C to 22 °C. Nonetheless, the tuber formation remained limited to the particular temperature (14 °C and 22 °C), and accelerated under short-day conditions [67,68]. It is now well understood that there is more shoot growth with more consumption of assimilates under high temperatures and, as a result, the tuber yield is drastically reduced [69,70]. In this context, physiologically efficient potato cultivars capable of early tuberization (even at relatively high temperatures and extended photoperiods) and a higher rate of photoassimilates being translocated to developing tubers may be beneficial. Variable photoperiods influence tuberization, which may be a micro-evolutionary indicator of the differential transduction of cell-to-cell signalling molecules under the spatial and temporal expression of regulatory genes involved in tuberization [71]. According to reports, the optimal conditions for potato tuberization are high irradiance, low temperature, and a short photoperiod. It is anticipated that regulatory genes encoding transcriptional activators or cell-to-cell communication molecules change more rapidly than structural genes. Increasingly, future studies will focus on identifying the putative regulatory elements in signal transduction pathways driving potato tuberization [72]. In the signal transduction pathway, the leaves perceive an adequate environmental cue, which is mediated by phyB and gibberellins, and then produce a systemic signal that is sent to the underground stolons to induce tuberization [73]. phyB-mediated perception of gibberellic acid (GA) response is controlled by a novel arm repeat photoperiod-responsive 1 protein (PHOR1), which is believed to be a general component of GA-signalling pathways [74,75]. Gibberellins and cytokinins are the two most important phytohormones that regulate the formation of potato tubers. GA also plays a function in the photoperiodic control of tuberization, and its endogenous levels are regulated by sucrose and abscisic acid (ABA). However, at high concentrations, it inhibits tuber induction in potato [76,77]. Efforts in this direction are necessary because they can compensate for the crop cycle being shortened due to high temperatures by utilizing the tropical region’s prevailing long photoperiod.

### 3.4. Tuber Development

Potato grown under long days with high temperatures may face delayed stolon and tuber initiation, with a reduction in the partitioning of photoassimilates to the developing tubers, which ultimately reduces the bulking (the size) and dry matter content (the quality) of tubers [12,78]. Potato is generally vulnerable to high-temperature stress, and the yield of the tuber is inversely proportional to the temperature. The optimal temperature for the growth and development of the haulm of potato plants is 20–25 °C, whereas the ideal temperature for tuberization and tuber growth is 15–20 °C [70]. The higher temperature during the potato growth stages leads to higher yield losses and more incidence of deterioration in the quality of tubers. High temperatures above 20 °C interfere with the partitioning of assimilates into the potato tubers, resulting in a low tuber yield. Lower tuber productivity under tropical regions with high temperatures leads to an increase in higher sugar retention in leaves, indicating that the translocation of photoassimilates is poor for the sinks (tubers) [12,79]. One of the reported reasons for this is the distorted phloem due to high temperatures, which adversely affects the source–sink relationship (mobilization of photoassimilates to tubers) [34]. Studies on tuber growth and development suggest that, under high temperatures, there is a reduction in the translocation of photoassimilates to the tubers and thus a declining production and productivity of potato. Accumulation of photoassimilates in the tuber tissues increases cell expansion and ultimately causes massive deposition of starch and storage proteins, thereby making strong storage sinks during the tuber development and bulking phases [80].

### 3.5. Photosynthesis

Many studies suggest a reduction in photosynthetic efficiency and tuber yield under high-temperature and drought stress [70,79,80]. In other studies, it was reported that temperatures of more than 30 °C completely inhibit photosynthesis in potato. Wahid [39] reported that photosynthesis is the most sensitive process under elevated temperatures. Under high temperatures, there is impairment in photosynthetic machinery, and other related physiological functions are often inhibited. The decrease in the photosynthesis rate caused by heat stress is associated with an increase in non-photorespiratory activities [39]. The key enzyme of carboxylation is RuBisCo, which is inhibited by high-temperature stress. Salvucci et al. [81] suggested that the reduction in the activity of RuBisCo under high-temperature stress might be due to the inhibition of Rubisco activation via a rapid and direct effect on RuBisCo activase (RCA). The heat stress leads to the denaturation and aggregation of the RCA protein, which further fails to activate Rubisco. Moreover, under heat stress, the fluidity of thylakoid increases, which leads to a reduction in the electron transport in the photosystem II (PS II) and its dislodging [78].

The investigations linking RuBisCo (ribulose-1,5-bisphosphatase) with photosynthesis reveal that, at elevated temperatures, soluble proteins such as RuBisCo and RuBisCo-binding proteins are reduced [82]. In leaves, the content of RuBisCo under elevated temperatures was found to correlate with the decrease in the photosynthetic rate of potato crop [47]. RuBisCo concentration was found to decrease under high temperatures supplemented by a corresponding decrease in its affinity for carbon dioxide. The effect of topicalization on potato could result in heat injury, protein synthesis suppression, and a loss of membrane integrity [83]. Under heat stress conditions, the respiration rate increases in proportion to the decrease in photosynthesis, and the combined result of these two processes is a decrease in net photosynthesis [84].

Carboxylation is catalyzed by ribulose bisphosphate carboxylase/oxygenase (RuBisCo), which can constitute up to 50% of the soluble protein in a leaf. Along with carboxylation, the RuBisCo protein is also involved in oxygenase activity. Heat stress can affect both carboxylation and oxygenation processes. High-temperature stress can enhance the oxygenase activity of RuBisCo, which leads to an increase in the production of H_2_O_2_, which is toxic to plant cells [85]. Meanwhile, photosynthesis was significantly affected under drought stress, mediated by the inactivity of stomatal and photosystem components [78]. Water-deficient conditions lead to a decrease in the quantum yield (Fv/Fm), electron transport rate, and photochemical quenching (Qp) [79]. In addition, the stomatal closure under water-deficient conditions reduces CO_2_ availability, which hampers the photosynthetic rate. The decrease in carboxylation efficiency and activity of RuBisCo was also reported under water stress conditions. Therefore, further study must be conducted with an emphasis on integrating features that can improve the efficacy of essential activities such as photosynthesis and respiration. This may be effective for the modulation of components that can also balance the source–sink features in order to achieve higher tuber yields with improved quality characteristics under tropical conditions [85].

### 3.6. Senescence

The leaf senescence of the plant was enhanced when exposed to heat, particularly at the time prior to or at the maturity stage of the plant, due to loss of chloroplastic integrity and chlorophyll synthesis, inhibition of PSII-mediated electron flow, and destruction of antenna pigments [86]. The recent report also suggested the same, where cultivars sensitive toward both high-temperature and water stress conditions show retarded sprout emergence, root growth, and low water potential, resulting in desiccation and early senescence [55].

### 3.7. Respiration

One of the critical characteristics affecting potato plant growth under heat stress conditions is mitochondrial respiration. When the temperature rises above the optimal temperature, gross photosynthesis is reduced, but normal respiration and photorespiration rates increase significantly [87,88]. The high respiration rate under elevated temperatures contributes to lowering potato yields [30,33]. Under high temperatures, the stored starch in the developing potato tuber is also utilized for respiration, contributing to a substantial decrease in the production and productivity of potato tubers [39,89]. Additionally, because of the high temperature, the rapid development pace results in a shortening of the entire growth cycle. In aggregate, the plant receives less time to collect photoassimilates, resulting in decreased yields [90].

### 3.8. Transpiration

Transpiration is another parameter that is strongly affected by heat and drought conditions in the potato plant. Due to rising temperatures, potato plants experience water stress as the plant’s transpiration rate increases, leading to greater demand for water from the soil. This raises the water requirement of potato crops [15]. If the water is not limiting, the high-temperature conditions result in a higher rate of CO_2_ assimilation, but this usually leads to a significant increase only in above-ground biomass production, with no net enhancement of photoassimilates portioned into the tubers [47]. The combination of heat and drought stress affects the transpiration rate significantly by affecting leaf area, root-to-shoot ratio, the orientation of the leaf, leaf thickness, leaf surface characteristics, and distribution of stomata on the leaf [55]. Growing potatoes in the tropical region will be challenging, where the plant will confer higher biomass production with lower tuber production due to improper allocation of photoassimilates.

## 4. Biochemical Changes and Responses of Potato Plants

The climate-related changes in temperate in subtropical and tropical regions have affected various biochemical events and processes in plants. The potato plant perceives abiotic stress such as heat and drought and responds dynamically by shifting its sugar and starch metabolism, ionic and osmolyte balance, synthesis of heat shock proteins, homeostasis of ROS, and other biochemical pathways [91,92,93]. Under heat and drought stress, the biochemical response includes accumulating reactive oxygen species (ROS), damage to cell membranes, electrolyte leakage, degradation of nucleic acids, and denaturation of proteins and enzymes [94]. The potato plant develops various strategies to mitigate the harmful effect of drought and heat stress to defend against damages caused by ROS and reactive nitrogen species [95]. The enzymatic (superoxide dismutase, ascorbate peroxidase, peroxidase, and catalase) and non-enzymatic (glutathione, ascorbate, polyphenols, vitamins, carotenoids) defence mechanisms effectively scavenge the ROS generated under heat and drought stress in potato [65,96]. Some of the key biochemical changes and responses in variable environments are as follows.

### 4.1. Carbohydrate Metabolism

The optimum temperature for photosynthesis and sugar and starch metabolism in potato leaves is about 24 °C [8]. Carbohydrate metabolism was reported to be severely affected by high-temperature stress, resembling one main condition in the tropical region [45]. During tuber development, sucrose produced by physiologically and photosynthetically active source leaves is transported to the developing tubers via the phloem [97]. The change in carbohydrate metabolism due to high-temperature exposure usually causes an increase in the levels of reducing sugars in the tubers. This ultimately affects the processing-related quality of potato tubers, as higher levels of reducing sugars adversely affect the chipping quality of processed potato products, even in processing-grade cultivars [98]. Starch synthase is the rate-limiting enzyme responsible for starch synthesis and its deposition in tubers [99]. Under high temperatures, the activity of the starch synthase enzyme decreases, resulting in a slower rate of starch deposition and a slower rate of tuber growth [99]. It was also reported that increased gibberellins due to high temperatures reduces starch synthase activity in developing tubers, preventing the sucrose from re-partitioning away from the tuber. Furthermore, enhanced levels of gibberellin synthesis due to high temperatures also influence tuber initiation and their further development [100].

During the nighttime, the plant’s sucrose is transported from the leaf and gets stored in sink tissue (tuber). Sucrose phosphate synthase (SPS) is the enzyme that controls photoassimilate partitioning by catalyzing the synthesis of sucrose, which ultimately contributes to the osmotically active driving force for phloem translocation [101]. SPS activity was increased under high temperatures in potato, whereas its activity is suppressed in tomato (>40 °C/25 °C, day/night) [53,102]. Metabolites are found to increase when potato is cultivated under high temperatures [103]. Under heat stress, hexose increases significantly in leaves and conversely decreases in tubers [47]. On the other hand, when sucrose and starch levels in the tuber were estimated, both metabolites were shown to decrease significantly in both the tuber and leaves of the potato plant under high-temperature stress. This suggests that the efficiency of sucrose-to-starch conversion in tubers is diminished as a result of decreasing sink strength [47]. Along with the metabolites, it was discovered that some hazardous compounds increased in concentration under high-temperature circumstances. For instance, the level of steroidal glycoalkaloid increased, resulting in bitterness in the tuber [104]. Plant breeders should consider this case when they develop potatoes for tropical climates.

### 4.2. Ionic and Osmolyte Balance

Potato plants grown in the tropical region are exposed to heat and drought stress, causing certain organic compounds of low molecular mass to accumulate. These compounds are referred to as compatible osmolytes [105,106]. In abiotic stress conditions, the potato plant accumulates various osmolytes such as sugar, sugar alcohol (mannitol and sorbitol), proline, and glycine betaine, which also confers tolerance to heat and drought stress [53,63]. They all help maintain the osmotic balance of the cell. The presence of compatible osmolytes helps cope with the high-temperature and drought stress by scavenging ROS generated due to stressful conditions [107]. Furthermore, these osmolytes also stabilize the protein structure, acting like molecular chaperones [108].

Glycine betaine is a soluble-compatible solute that plays an essential role in tolerance against high-temperature stress [109]. A higher concentration of glycine betaine was reported in a heat-tolerant cultivar of potato (Kufri Surya) compared to susceptible cultivars such as Kufri Chipsona 3 [53]. Accumulation of proline leads to protein stabilization. Due to its metal chelator properties, it acts as a molecular chaperone or chemical protein chaperone. It also acts as an antioxidative defense molecule that scavenges reactive oxygen species (ROS) and has signaling behavior to activate specific gene functions that are crucial for plant recovery from abiotic stresses [110,111]. Additionally, it acts as a supply of carbon, nitrogen, and energy during periods of water-deficient conditions [49]. As a result, biosynthesis of osmolytes is controlled, especially during times of stress, so that they can help plants grow and develop in the tropical climate.

### 4.3. Heat Shock Proteins (HSPs)

Heat shock proteins (HSPs) are major proteins which act as molecular chaperones and are the key components responsible for protein folding, assembly, translocation, and degradation under abiotic stress conditions [112]. As chaperones, these proteins prevent the irreversible aggregation of other proteins and play a role in the refolding of proteins under heat stress conditions [113]. The synthesis and accumulation of HSPs under heat stress in the plant have been shown to provide tolerance against high-temperature stress [114]. HSPs are categorized into six different families based on their molecular weight and sequence homology, which include Hsp110 (>100 kDa), Hsp100 (90–100 kDa), Hsp90 (80–90 kDa), Hsp70 (66–78 kDa), Hsp60 (50–60 kDa), and Hsp20 (15–39 kDa) [115]. Hsp20s in potatoes (*StHsp20*) are sensitive and positively regulated under heat stress and they provide thermotolerance to potato plants [50]. Hsp90s are reported as a highly conserved molecular chaperone among all the HSPs, where it is distributed in the cytoplasm, chloroplasts, and mitochondria [116]. Earlier reports revealed that heat shock cognate 70 (HSc70) expression was increased in the Désirée cultivar of potato and positively correlated with the improved tolerance and tuber yield in heat stress conditions [51]. Savić et al. [117] suggested that electrolyte leakage assay in combination with immunoblot measurement of HSP accumulation under heat stress conditions could be a reliable method for screening potato genotypes for heat tolerance and identifying heat-tolerant potato cultivars. Other reports also suggest that the exogenous application of phytohormones such as salicylic acid in potato microplants showed a higher expression of HSPs that provide tolerance to high-temperature stress [118]. Thus, HSPs can play a significant role in maintaining the growth and development of potato plants and tubers under tropical and subtropical conditions. Therefore, this area holds promising aspects with respect to the ability of potato plants to cope with high-temperature stress.

### 4.4. ROS, RNS, RSS, and Antioxidant System

Under heat stress, reactive oxygen species (ROS) accumulate, which causes severe oxidative damage to the plant, thus inhibiting growth and development-related activities [119]. When ROS production increases more than the cellular scavenging capacity, there is an imbalance in redox homeostasis, resulting in more membrane damage (Figure 2). Consequently, more electrolytes are leaked and disrupt the functioning of the cell [40]. Besides the ROS, there are other reactive molecules which originate from other elements, such as nitrogen and sulphur, and they are referred to as reactive nitrogen species (RNS) [120] and reactive sulphur species (RSS) [121]. Overall, the reactive species viz. ROS, RNS, and RSS cause oxidative stress-like conditions in the plant or plant part.

The ROS include hydroperoxyl radical (HO_2_^•^), superoxide anion (O_2_^•−^), alkoxy radical (RO^•^), and hydroxyl radical (•OH), and also non-radical molecules such as hydrogen peroxide (H_2_O_2_) and singlet oxygen(^1^O_2_). Likewise, RNS include nitric oxide (NO) [120] and RSS include thiol, disulphide, sulfenic acid, thiosulfinate, and thiosulfonate [122]. They also act as potential signalling molecules under oxidative stress conditions. The enzymatic components are involved in scavenging different types of reactive species formed in the plant under drought and heat stress conditions [120,123]. The antioxidant enzymes such as SOD, CAT, and APX were reported to increase significantly under heat and drought stress conditions (Table 1). The potato plant can cope with the increased levels of reactive species in the cellular system [55,124]. In addition to this, it was reported by Arora et al. [125] that O_2_^•−^ can react with NO, and this leads to the production of peroxynitrite (ONOO^–^), which is a powerful oxidant and is involved in post-translational modification of protein through tyrosine nitration. Researchers need to decipher whether the mitigation of oxidative stress is caused due to ROS, RNS, and RSS in potato grown in non-conventional tropical areas. This strategy will help design potato plants that are more suitable and adaptable to the new environment.

## 5. Approaches for Adaptation of Potato Plant

Potato plants have insufficient abiotic stress tolerance due to the limited genetic diversity of present potato cultivars [126]. Thus, in order to withstand varied environmental stressors, the current commercial cultivars must be modified accordingly. However, little is known about the distinct molecular pathways involved in potato tuberization in response to heat stress. Understanding the role and network of different genes involved in heat stress resistance requires a full characterization from an agrobiotechnology standpoint. Under a changing climate, sustainable potato production necessitates an interdisciplinary strategy that includes innovative technologies, molecular biology, agronomy, breeding, and agrometeorology.

### 5.1. Agrometeorology-Based Crop Modelling and Agronomic Practices

With the advances in climatology and information technology, it is now simple to analyze genotypes by environmental interactions and create crop models [127]. A comprehensive model of the growth and development of potatoes called LINTUL-POTATO was published in 1994. The mechanistic model recreated the early stages of crop development, such as emergence and leaf growth, as well as the process of light absorption, which continued until extinction [128]. The aforementioned model was enhanced and improved with unique computations to investigate tuber quality features such as tuber size distribution and dry matter concentration in relation to crop environment and management, known as LINTUL-POTATO-DSS [127]. Statistical analysis and records of weather trends may be able to assist us in locating new places that are ideally suited for potato cultivation. Similarly, the DSSAT SUBSTOR-Potato model [129] was used to simulate potato tuber yield in Indian conditions, suggesting that, if planting is postponed beyond November, all of the cultivars in the experiment (Kufri Jyoti, Kufri Pukhraj, and Kufri Himalini) are likely to see a significant decrease in tuber production [130,131]. Out of these cultivars, Kufri Pukhraj was projected to remain a viable cultivar under subtropical conditions until 2050 and can be planted until the first week of December. Modulating potato planting time by using different cropping models might be a strategy to understand the exact date of sowing in tropical areas.

Potato cultivation under changing climatic scenarios requires improved potato cultivation techniques. Appropriate agronomic practices such as planting date, soil management, tillage, mulching, irrigation, intercropping, and superior genotypes tolerant to high temperature and drought can contribute to higher yields [49,132]. Aeration, infiltration, and nutrient uptake are all things that can be improved by tilling the soil. The soil’s moisture and temperature are directly related to the yield and quality of potatoes, and these factors can be controlled by the use of organic and biodegradable mulches [133]. Rykaczewska [134] demonstrated that two weeks of heat and drought stress during the flowering phase of potato plants can diminish their yield by more than 35%, which is also a factor for the development of secondary tuberization. Due to the shallow nature of the root architecture, potato plants are susceptible to drought [135]. Therefore, improved irrigation techniques such as drip irrigation can increase potato output and quality traits under abiotic stress. Moreover, plants exposed to moderate temperatures over a brief period of time can build a memory for stress tolerance, allowing them to live at temperatures where normal plants cannot, this process is known as acquired thermotolerance [51]. The acclimatization of potato plants to 25 °C for two hours is connected with changes in the expression of several heat shock proteins, modifications to the cell wall, abnormalities in hormonal signalling, and chromatin remodelling. Later, when these plants were exposed to 40 °C, their growth was superior to that of control plants [51]. In addition to the adaptation, which includes the development of heat-tolerant cultivars, adjusting plant and harvesting time might help to shift the location of where potato production will be higher. In certain tropical highland regions, potato growing is limited to higher zones (Puna and Pramo zones of Andes; Nilgiri hills of southern India) [27,136].

### 5.2. Exploring Genetic Diversity and Breeding

Due to the fact that thermotolerance is a multigenic trait, it is imperative that appropriate methods be utilized in order to evaluate the genetic diversity in both inherent and acquired tolerance [136,137]. Due to their resilience to pests and diseases and their ability to adapt to harsh climates, wild potato species are of great interest to potato breeders [138]. Wild potato species inhabit a wide variety of settings and could be anticipated to have varying genetic degrees of stress tolerance [137]. Several wild potato species can withstand high temperatures, including *S. kurtzianum*, *S. sogarandinum*, *S. chacoense*, *S. stoloniferum*, *S. demissum*, and *S. berthaultii*, and these wild potato species can be utilized in breeding programs to create heat-tolerant lines [139]. The selection of superior phenotypes or parents is the most important phase in breeding schemes for crop improvement. According to Rykaczewska [134], the impacts of heat stress are more detrimental during the early growth phases. Nevertheless, with the recurrent method of selection, the frequency of desirable alleles can be increased by selecting the superior genotypes over the base population. Knowledge of the molecular pathways involved in heat tolerance can serve as a foundation for producing high-yielding, stress-tolerant cultivars through molecular breeding, transgenic, and genome-editing techniques.

### 5.3. Molecular and Transgenic Approaches

A complex biochemical pathway and gene network contributes to the plant’s tolerance to drought and heat stress. Most significantly, tuberization is impaired considerably in potatoes when heat and drought stress are combined, which is mediated by various transcription factors [140]. The molecular network of various transcription factors, hormones, and signalling molecules is involved in coping with climatic conditions and changes when potato plants grow under tropical conditions [55]. The increased expression of starch-degrading genes under drought and heat stress is one of the detrimental factors involved in the degradation of potato tuber quality in terms of carbohydrate metabolism [46]. The tuberization signal in potato is highly dependent on StSP6A, an orthologue of the Arabidopsis protein FLOWERING LOCUS T (FT). StSP6A is highly regulated by elevated temperatures, and it also affects the accumulation of photoassimilates in the tuber in the form of starch [141]. The tropicalization of potato will involve various interactions, including at the gene level. The future implications of the interactions that can assist in coping with the harsh environment and resist the negative impact of such an environment need the attention of researchers.

The transgenics approach has been used to develop tolerance against drought and heat stress [40]. The genes such as C-repeat Binding Factors/Dehydration-responsive element binding (*CBF*/*DREB*) play an essential role in signal transduction and gene expression under heat and drought stress in potato plants [52]. They reported that overexpression of the *ScCBFI* gene in transgenic *Solanum tuberosum* from *Solanum commersonii* results in better root development and plant growth under water-deficient conditions (Table 2). Targeting such genes in potato plants can prove beneficial in terms of making the potato plant suitable for growing in tropical and subtropical regions with minimum effect on its production and productivity. Other reports on transgenic potatoes showed an increase in the level of ascorbate in the plant, providing tolerance against water and salt stress. The tolerance was achieved by overexpression of the *Arabidopsis thaliana* Dehydroascorbate Reductase gene (*AtDHAR1*) in the transgenic potato plant, with 4.5 times more DHAR activity in comparison to the wild type [142].

As discussed earlier in this section, SP6A is an essential gene that is involved in temperature-mediated tuberization in potato. The favorable condition triggers the process of tuberization by perceiving the signal (StSP6A) from leaves and transporting it to stolons. The tuberization was reported to be suppressed when StSP6A was silenced using RNAi-based downregulation of this gene, while its upregulation promoted the tuberization [141]. Kloosterman et al. [148] identified CYCLING DOF FACTOR 1 (*CDF1*), a transcription factor required for the maturity and initiation of tuber development under long-day conditions. *CDF1* was found to involve direct regulation of SP6A expression. An illustration related to the regulation of the SP6A gene is presented and described in Figure 3.

TFs such as *WRKY*, *AP2*, *NAC*, *DREB*, *bZIP*, and *BELL* are involved in the activation/inactivation of various metabolic pathways, plant hormones, and transcriptional and post-transcriptional regulation under heat and drought stress (Figure 2; Table 2). In potato plants, 70 and 21 TFs were differentially expressed after short (6 h) and prolonged (3 days) exposure to heat, respectively [103]. Under short heat exposure, 14 TFs were upregulated and 56 TFs were downregulated significantly, whereas prolonged heat exposure to potato plants showed upregulation and downregulation of 7 and 14 TFs, respectively. The analysis of TFs and the network of the genes in potato helps to gain insight into the molecular level, working in terms of mechanisms and responses under heat and drought stress conditions. These parameters mentioned above help to provide an essential knowledge base for designing potato breeding programs and strategies suitable for tropical and other non-conventional areas on the globe.

### 5.4. CRISPR Approach for Molecular Insight and Functional Gene Analysis

Plant genome editing (GE) has advanced rapidly in recent years, opening up new opportunities and intriguing possibilities for both fundamental research and plant breeding. The type II CRISPR/Cas9 (clustered regularly interspaced short palindromic repeats and CRISPR-associated protein 9)-mediated GE system from *Streptococcus pyogenes* has been widely used by plant scientists. It consists of two parts: the RNA-guided DNA endonuclease *Sp*Cas9 and a single-guide RNA (sgRNA) [149]. It looks for a 5′-NGG-3′ PAM (protospacer-adjacent motif) in the genome, which triggers melting of nearby DNA and a search for a spacer sequence that matches the 5′ end of a sgRNA. This results in a double-stranded DNA break (DSB) about 3bp upstream of the PAM, which is caused by the combined activity of the RuvC and HNH endonuclease domains [150]. NHEJ (Non-homologous end joining) DNA repair is activated upon a DSB, which can lead to small indels at the breaking site, which results in gene knockout through frameshift mutations [151]. While most research has focused on the generation of loss-of-function alleles, novel CRISPR tools, such as the CRISPR-mediated base-editing system, which allow precise base conversion without the need for donor DNA or the development of a DSB, have recently been created [152]. There have so far been two types of base editors (BEs) discovered and developed: adenine base editors (ABEs) and cytosine base editors (CBEs). Both of these types of base editors (BEs) are composed of an RNA-independent Cas9 with impaired DNA cleavage activity, typically nCas9 (nickase Cas9) for plant crop applications, and a catalytic domain associated with adenine or cytosine deamination, respectively [153,154].

GE has a lot of potential for improving crops, but it has not been fully realized in clonally propagated tetraploids such as the potato, despite its widespread use. In potatoes, CRISPR/Cas experiments have increased tuber starch quality, carotenoid biosynthesis, glycoalkaloids, and enzymatic browning [155,156,157]. Functional mutants were generated to investigate phenotypic and herbicide tolerance differences [158]. Furthermore, the efficacy of Cas9 prime editing and base-editing tools for tolerance to herbicides in potato has been successfully established by researchers [159,160]. Additionally, self-compatible regenerants were generated via *Agrobacterium* or virus-induced genome editing using Cas9 (VIGE) [161,162]. However, there are several challenges to CRISPR implementation in potato.

Only a few potato cultivars are responsive to transformation; others must be examined in tissue culture for regeneration and transformation. Protoplast transformation and regeneration from leaf protoplasts might result in somaclonal variation, which can negatively influence crop growth and development. Transformation techniques such as agrobacterium, biolistic or particle bombardment, and the floral-dip method have been used regularly in potato [163]. For CRISPR/Cas in potato, sgRNA dicot-origin promoters such as Arabidopsis (*AtUp*)/potato (*StU6p*)/U3p and plant promoters such as *CaMV 35S* are the most often used in protoplasts and *Agrobacterium*-mediated transformation [164]. Furthermore, the genetically complex tetraploid potato crop, on the other hand, makes it challenging to utilize the *A. tumefaciens*-mediated technique to deliver ribonucleoprotein (RNP) complexes and to remove Cas9 from the crop genome through backcrossing or selfing [165]. Potato breeders are actively striving to re-invent the crop to speed up progress in understanding the complex genetic characteristics such as quality, yield, and tolerance to stress. CRISPR/Cas-mediated GE will be a game changer in terms of genetic improvement, opening up new opportunities for developing a stronger potato breeding pipeline.

## 6. Conclusions

The potato plant is vulnerable to high-temperature and water-deficient conditions. Therefore, the high-temperature and drought stress in potato growth and development lead to alterations in morphological, physiological, biochemical, and molecular responses. Tuberization is one physiological process that is adversely affected by high-temperature and water-deficient conditions. In addition, there is a change in shoot growth, reduction in the root-to-shoot ratio, and increase in the incident of senescence. The biochemical changes in potato plants under tropical conditions include accumulation of osmolytes such as proline and glycine betaine, decreased starch accumulation in tubers, increased reducing sugar content, synthesis of HSPs, generation of ROS, and activation of various protective and scavenging mechanisms. The network of genes and TFs are involved in response to stressful conditions at the molecular level. Tuberization is a complex multigenic trait induced under short-day and low-temperature conditions via the interactions of various hormones, genes, and TFs. Various TFs such as *WRKY*, *NAC*, *DREB*, *ERF*, and others play a role in coping with the heat and drought stress. This role of the StSP6A gene is very important, as it is responsible for tuberization. Moreover, its expression is temperature-dependent. These aforementioned traits and genes need to be explored by physiologists and breeders to develop tolerant cultivars of potato. The selection of early maturing germplasm is also crucial in breeding programs of potato for short-day conditions. The use of agrometeorology-based crop modelling and agronomic practices might be useful under climate change and for particular areas where the potato crops are grown. Exploring several wild potato species and their genetic diversity with regard to which can withstand high temperatures and drought can be utilized in breeding programs to create heat and drought-tolerant lines. Modern genomic tools to develop climate-resilient, transgenic, and non-transgenic cultivars might be employed for enhanced drought and heat stress tolerance. Various approaches and their integration are the best possible way to pave the way forward to enhance the production and productivity in the developing nations of tropical regions. These strategies will also be a step forward in fulfilling global food and nutritional security.

## Figures and Tables

**Figure 1 plants-11-02857-f001:**
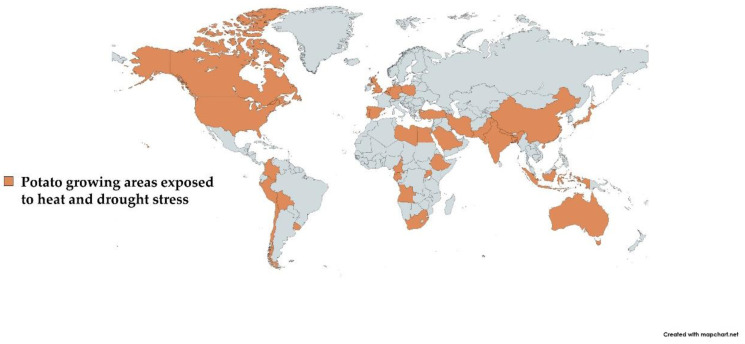
The area shown in brown color depicts the major potato-growing areas affected due to heat and drought stress.

**Figure 2 plants-11-02857-f002:**
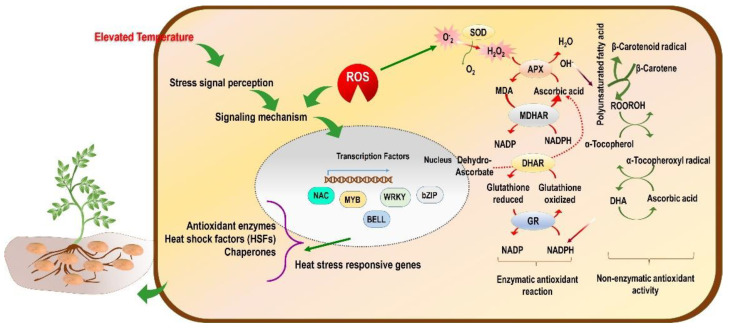
Conceptual model of the effect of high-temperature stress (tropicalization) on molecular mechanisms and ROS-mediated regulation of enzymatic and non-enzymatic antioxidant activity in potato plants.

**Figure 3 plants-11-02857-f003:**
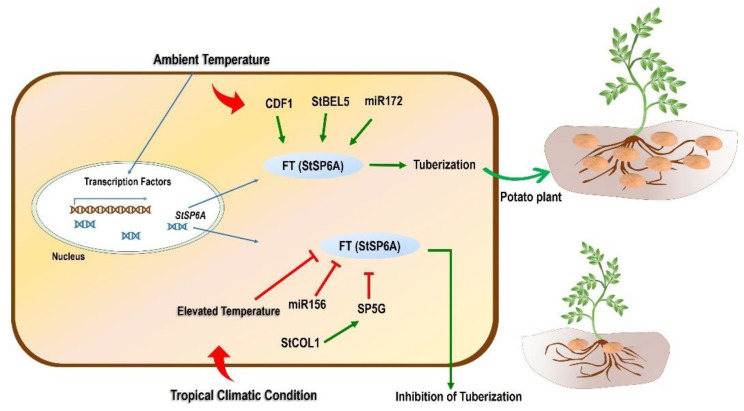
Schematic model of the effect of tropicalization on tuberization. Temperature plays an important role in the expression of essential genes responsible for tuberization. The master regulator for tuberization in potato is the FT(StSP6A) gene responsible for tuberization under normal/ambient temperature conditions. Different TFs and miRNA also control the process of tuberization through an StSP6A-mediated mechanism. TFs such as CDF1 and StBEL5 upregulate the expression of StSP6A, whereas miR172 also upregulates this gene, leading to tuberization. However, when the potato is grown in tropical conditions and is subjected to high temperatures, tuberization is inhibited. An FT family member, SP5G downregulates the expression of the StSP6A gene. The former SP5G is reported to be regulated positively by StCOL1 expression. Moreover, miR156 also negatively controls the expression of StSP6A, which ultimately inhibits tuberization. CONSTANS-LIKE protein 1: StCOL1; CYCLING DOF FACTOR 1: StCDF1; BELLRINGER-1 LIKE 5: StBEL5.

**Table 1 plants-11-02857-t001:** Physiological and biochemical responses of potato under heat/ drought stress.

Trait	Response to Tropicalization (Heat/Drought Stress)	Reference
Yield and total biomass production	Reduction in both yield and biomass	[46]
Root growth	Root growth was stimulated under drought stress	[46]
Photosynthesis	Reduction in the PSII efficiency and ultimately reduction in photosynthesis	[40,47]
Clorophyll and carotenoid content of leaves and tubers	Significant reduction in both chlorophyll and carotenoid content	[48]
Membrane stability index (MSI) and cell membrane stability (CMS)	Reduction in membrane stability index under heat and drought stress	[40,48]
Relative water content (RWC)	Reduced under both heat and water-deficient conditions	[40]
Water potential	Significant reduction in water potential	[49]
Water use efficiency (WUE)	Reduction in WUE mainly under drought conditions	[49]
Tuber bulking	Reduce dry matter partitioning, reduce tuber filing, increase the production of secondary tubers, increased russeting and cracking	[21]
Tuberization	High temperature reduces tuberization	[16]
HSP20	ATP-independent molecular chaperones inhibit the irreversible aggregation of denaturing proteins, thus enhancing the thermotolerance of the plant	[50]
Starch synthesis	Heat leads to the production of reactive oxygen species that interfere with starch synthesis	[51]
Sucrose synthesis	A decline in sucrose content	[47]
Proline content	Acts as osmotic agent, protecting plant cells from dehydration	[52]
Glycine betaine	Acts as osmoprotectant in potato under heat and drought stress	[53]
Sucrose synthase	Degradation of sucrose into hexoses	[47]
Superoxide dismutase	Scavenges superoxide molecule under stress conditions and prevents oxidative damage in potato	[53]
Ascorbate peroxidase	Activity increased under heat and drought stress and provides tolerance in potato	[53]

**Table 2 plants-11-02857-t002:** Role of the different genes influencing various traits during the process of subtropicalization.

Genes Expressed	Role of Gene and Molecular Response	Reference
*DHAR1*	Synthesis of ascorbic acid, which acts as a strong antioxidant, protecting chlorophyll against degradation, and allowing faster removal of H_2_O_2_	[142]
*DREB1B*	TF involved in enhancing drought tolerance	[49]
*ScCBF1*	The CBFs bind to the cold/dehydration-responsive regulatory motif (CRT/DRE) and are present in the promoter of many drought and cold-responsive genes, such as those associated with osmoprotectant, cold-responsive (COR), and late embryogenesis-abundant (LEA) proteins	[52]
*HSPs*	Molecular chaperones prevent denaturing and aggregation of proteins under heat stress	[47]
*SAM-DC*	Mediates the polyamines, spermidine, and spermine involved in providing tolerance against stress	[143]
*StSP6A*	Master regulator of tuberization and is significantly affected by temperature	[16,47,144]
*StCOL1*	Suppresses tuber formation by activation of FT-like StSP5G repressor	[145]
*ABRE* and *DRE*/*CRT*	The promoters of drought-tolerant gene. ABRE is a major cis-acting element in ABA-responsive gene expression.	[146]
*LEA5*	Provides tolerance against drought stress	[46]
*MYB96*	Involved in ABA and auxin cross-talk, increasing lateral root formation	[147]

DHAR, Dehydroascorbate reductase; ScCBF1, *Solanum commersonii* C-repeat Binding Factors; SAM-DC, S-adenosylmethionine decarboxylases; StCOL1, StCONSTANS-like1; FT, FLOWERING LOCUS T; CO, CONSTANS; ABRE, ABA-responsive element; DRE, Dehydration-responsive element; CRT, C-RepeaT; LEA, Late Embryogenesis-Abundant.

## Data Availability

Data sharing is not applicable as no new data were generated or analyzed during this study.

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
