# Peer review of "Mechanistic Concept of Physiological, Biochemical, and Molecular Responses of the Potato Crop to Heat and Drought Stress"

_plants, 2022, doi:10.3390/plants11212857_

Round 1

Reviewer 1 Report

It's a very well-written and timely review of potato growth responses to heat and drought stress. In enjoyed reading it and I fully share the concerns of authors. Potato production will be severely affected by the higher temperatures and more variable precipitation and thus drought. Therefore action is urgently needed with respect to breeding more tolerant varieties. I have a number of minor suggestions and edits:

Abstract: Please indicate optimum temperature range for avoiding heat stress

Line 76: Insecurity?

Line 91: Reference should be a number (17)

Line 118 example of heat impact in temperate climate showing 25-30% yield loss at present due to heat stress: Peng, J., Manevski, K., Kørup, K., Larsen, R., Zhou, Z., Andersen, M.N. 2021. Environmental constraints to net primary productivity at northern latitudes: A study across scales of radiation interception and biomass production of potato. International Journal of Applied Earth Observations and Geoinformation. 94, 102232. https://doi.org/10.1016/j.jag.2020.102232

Line 155: Field investigations shows that the optimum daily mean temperature may be as low as 13 oC. see previous reference

Line 198-199: In the tropical climate, high temperature leads to the development ...

Line 203: The leaf is the primary source of the potato plant's assimilates, ...

Line 206: extensively

Line 208: the globalization of potato was achieved by

Line 220: It's not clear from the figure text or the text here what is meant by "temperature requirement". They seem very high. Does it mean maximum temperature that allow development? Clarification is needed.

Line 255: "is the best crop for responding to high temperatures, rising CO2 levels, and nutritional stress in one way [65]" It's not clear what is meant by this statement. In which way?

General comment: It is mentioned in the article that phenotyping is needed but I'm lacking a short paragraph on possible avenues to accomplish this in the section concerning "Approaches for adaptation of potato plant". phenotyping could be done both at lab, greenhouse and field scale, using some of the many new technologies both in molecular biology and at lab and field scale new sensing technologies e.g. thermal cameras mounted on drones.

Author Response

Reviewer #1

Comment: It's a very well-written and timely review of potato growth responses to heat and drought stress. In enjoyed reading it and I fully share the concerns of authors. Potato production will be severely affected by the higher temperatures and more variable precipitation and thus drought. Therefore, action is urgently needed with respect to breeding more tolerant varieties. I have a number of minor suggestions and edits:

Response: Authors sincerely acknowledge the reviewer's valuable positive and constructive remarks and positive appreciation. The suggestions have been well taken and incorporated into the manuscript in track change mode.

Comment: Abstract: Please indicate optimum temperature range for avoiding heat stress

Response: The suggestion is incorporated in line 28-29

Comment:Line 76: Insecurity?

Response: The error is rectified in line 77.

Comment:Line 91: Reference should be a number (17)

Response: The refence is corrected.

Comment: Line 118 example of heat impact in temperate climate showing 25-30% yield loss at present due to heat stress: Peng, J., Manevski, K., Kørup, K., Larsen, R., Zhou, Z., Andersen, M.N. 2021. Environmental constraints to net primary productivity at northern latitudes: A study across scales of radiation interception and biomass production of potato. International Journal of Applied Earth Observations and Geoinformation. 94, 102232. https://doi.org/10.1016/j.jag.2020.102232

Response: The example is incorporated in the text (Line 118).

Comment: Line 155: Field investigations shows that the optimum daily mean temperature may be as low as 13˚C. See previous reference.

Response: As per suggestion we have now modified the line 169-170.

Comment: Line 198-199: In the tropical climate, high temperature leads to the development ...

Response: We are thankful and we have rectified the error.

Comment: Line 203: The leaf is the primary source of the potato plant's assimilates, ...

Response: We are thankful and we have rectified the error.

Comment: Line 206: extensively

Response: The error is corrected.

Comment: Line 208: the globalization of potato was achieved by

Response: The error is corrected.

Comment: Line 220: It's not clear from the figure text or the text here what is meant by "temperature requirement". They seem very high. Does it mean maximum temperature that allow development? Clarification is needed.

Response: As per the suggestion of the both the reviewers we have now removed the figure 1 from the text. We found the information provided was the repetition of the previously published data so we apologize the reviewer for this.

Dutt, S.; Manjul, A.S.; Raigond, P.; Singh, B.; Siddappa, S.; Bhardwaj, V.; Kawar, P.G.; Patil, V.U.; Kardile, H.B. Key players associated with tuberization in potato: potential candidates for genetic engineering. Crit. Rev. Biotechnol. 2017, 37, 942–957.

Comment: Line 255: "is the best crop for responding to high temperatures, rising CO2 levels, and nutritional stress in one way [65]" It's not clear what is meant by this statement. In which way?

Response: We have removed the present sentence due to lack of clarity.

Reviewer 2 Report

Review

Tropicalization of Potato: Dynamics of physiological, biochem-

ical and molecular responses under heat and drought stress

The manuscript reviews the effect of heat and drought on tuber productivity, plant and tuber morphology, physiological processes, biochemical composition, and molecular genetics. It is a good review of the topics but does not provide much advice about germplasm screening and sources of heat and drought tolerance in cultivated potatoes.

The title emphasizes ‘tropicalization’; however, tropicalization is not well defined. In addition, temperate climates also have issues with heat and drought (i.e., heat in Europe and US in recent years). Thus, ‘tropicalization of potato’ should be removed from the title, and ‘tropicalization’ should be revisited in the whole manuscript.

Some original data or summaries should be included. For example, include maps illustrating regions in the world the authors are targeting (the review should expand beyond India; what about Africa, South America?). Summarize results from surveys about cultivated potato varieties that perform well in the targeted areas and add results from surveys from researchers in target areas about screening approaches to screen for heat and drought tolerance. Include a table showing selected areas where the highest yields are obtained and speculate why. Talk more deeply about yield losses due to heat and drought everywhere in the world.

The role of photoperiod on tuberization should be discussed in the context of expanding potato production to tropical areas (and various latitudes).

Review the format used for references. Genus and species should be in italics; page numbers are missing in several references; inconsistent use of bold font…

Request review of English language and grammar.

Below are some suggested edits, but the whole manuscript should be reviewed in depth.

Line 33: Replace ‘maintenance’ with ‘maintaining’

Line 37: Replace ‘evaluation’

Line 38 Replace ‘response’ with ‘responses’

Line 38-39: Replace ‘a prominent factor’ with ‘prominent factors’

Line 41: Replace ‘these regions’ with ‘tropical regions’

Line 43: All words included in the title are considered keywords. Thus, do not repeat them. Use different ones. For example, the scientific name of cultivated potato, heat tolerance, drought tolerance, climate change

Line 45: Move the introduction to the next line

Line 46: If you refer to the cultivated tetraploid potato, you should say Solanum tuberosum spp. tuberosum L.

Line 46: Replace ‘crop’ with ‘food crop’. Potato would rank four if corn is considered, Corn is mainly used for feed, thus if you indicate ‘food crop’ potato ranks third.

Line 48: Indicate in parenthesis the year associated with the production year mentioned.

Line 48: The increase will not be everywhere. In fact, in some areas it will decrease. Indicate in what areas production will increase (‘in several regions’)

Line 49: Replace ‘well-known’ with ‘popular’

Line 49: Replace ‘Indians’ with ‘many countries’

Line 49: What about other countries like China, Russia, and the USA?

Line 52: Replace ‘it’ with ‘potato’

Line 53: Replace ‘this crop’ with ‘the potato crop’

Line 56: Replace ‘the aforementioned issue is also the’ with ‘climate change will become a ’

Line 58: Domestication is the process of going from wild to domesticated. Thus, remove ‘cultivation’

Line 58: Replace ‘was done’ with ‘took place’

Line 58: Replace ‘hills’ with ‘highlands’

Line 60: Remove ‘men’

Line 61: Is this statement applied worldwide? If so, start by saying ‘Worldwide’ (line 60)

Lines 61 to 63. The use of ‘summer crop’ and ‘winter crop’ seems to be different than in the USA. In the USA, we use ‘fall crop’ if the crop is harvested I the fall and ‘summer crop’ if the crop is harvested in the fall. I believe you are using different terminology. To be consistent, indicate when the crop grows and is harvested. Refer to national USA statistics to see how fall and summer crops are reported. Most of the production in the USA is considered a fall Crop.

Line 64: Replace ‘as a winter crop’ with ‘during the winter’

Line 65: Review references for temperature for tuberization, temperatures below 20 are usually listed as required for tuberization. Temperatures above 20 at night reduce tuberization. You could end the phrase at below 22 C and start a new phrase saying that temperatures below 20 C are usually required for tuberization and list some references.

Line 72: Remove ‘that’

Line 74: Replace ‘recent past’ with ‘recent years’

Line 74: Define ‘tropicalization’ in one phrase.

Line 75: Expand to indicate where. ‘in tropical and sub-tropical countries’?

Line 75: Add ‘crop’ after ‘potato’

Line 76: Replace ‘achieve better’ with ‘contribute to improve’

Line 77: Replace ‘cues like’ with ‘factors such as

Line 77: Add ‘flooding’?

Line 78: Add comma before ‘and productivity’.

Line 82: Add comma before ‘and’

Line 92: Replace ‘a staple food’ with ‘staple foods’

Line 95: Replace ‘response’ with ‘responses’

Line 96: Replace ‘towards’ with ‘to’

Line 96: Remove ‘and implication of knowledge’

Line 96: Replace ‘mechanism’ with ‘mechanisms’

Line 97: Replace ‘response’ with ‘response to abiotic stresses’

Line 98: Replace ‘by the introduction of tolerance’ with ‘with the goal of developing tolerant varieties

Line 100: Replace ‘a detrimental effect’ with ‘detrimental effects’

Line 103. Rephrase. The potato crop originated in borderline subtropical/alpine climate and performs best places with warm days and cool nights. Potato production in tropical and sub-tropical regions is challenging

Line 104: Replace ‘the tropical region’ with ‘tropical regions’

Line 107: Remove ‘Presently,

Line 108: Add ‘In addition, heat and drought spells are becoming more frequent in temperate zones’. Add references.

Lines 111-112. Remove ‘prevail in tropical areas’

Line 116: Remove ‘Unlike many crop species’

Line 122: Remove ‘The’

Line 123-124. Add period after ‘changes’

Line 124: Remove ‘in their’

Line 124: Start new phrase with ‘Growth…

Line 124: Add ‘seriously affected’ after ‘development’

Line 126 Replace ‘feature’ with ‘features’

Line 131: Replace ‘summer’ with ‘the summer’

Line 131: Remove ‘The’

Line 132: Replace ‘imbalance’ with ‘imbalances

Line 144: Replace ‘episode’ with ‘treatment’. Remove ‘treatment of’

Line 147: Rewrite. Planting time affects potato yield.

Line 147: Indicate if Spring planting typically results in higher yields.

Line 148: Replace ‘The high temperature’ with ‘High temperatures’ …

Line 149: Replace ‘is’ with ‘are’

Line 149: Add ‘combination’ before ‘low’

Line 150: Remove ‘The’

Line 153: Replace ‘the high’ with ‘high’

Line 153: Replace ‘the potato’ with ‘potato’

Line 158: Replace ‘the biomass’ with ‘biomass’

Line 162: Plant growth differences due to growth under high temperature stress would result in differences in the size of xylem and phloem but cause and effect in relation to temperature can not be oversimplified.

Line 166: Remove ‘the’ in ‘the temperate’

Line 167: Remove ‘the’ in ‘the physiological’

Line 167: Do you mean ‘now adopted’ instead of ‘not adapted’?

Line 168: Replace ‘the potato plant’ with ‘potato plants.

Line 169: Replace ‘adapt’ with ‘acclimate’. Adapt is an evolutionary term.

Line 170: Replace ‘on the sensitivity’ with ‘on the intensity’

Line 170: Remove: of that is exposed’

Line 172: Explain the meaning of size (mass?)

Line 175: night temperatures are more important for stolon formation. Indicate average day and also night temperatures.

Line 176: Do you refer to day or night temperatures? Above ground, growth is good above 22 C

Line 177: Explain the meaning of reproductive growth. Do you mean tuber enlargement?

Line 178: Replace ‘against’ with ‘to’

Line 180: Replace ‘The other’ with ‘Other’

Line 181: Mention in parenthesis the anatomical features you are referring to.

Line 182: Explain extrinsic and intrinsic.

Table 1. Table heading. Remove ‘Different’

Table 1. Qualify biomass. Do you referent to above-ground plant biomass?)

Table 1: Reduction of carotenoids refers to the plants? or the tubers?

Table 1: Reference 21. Add ‘reduce’ before tuber filing. Replace ‘increase secondary tuber’ with ‘increase the production of secondary tubers’ Does russeting and cracking increase?

Table 1. Reference 47. Remove the first comma.

Table 1. Reference 48. Replace ‘which’ with ‘that’

Table 1. Reference 41. Indicate at what level there is reduction of sucrose?

Table 1. Reference 49. Increase in proline content? Replace ‘act’ with ‘acts’ and ‘agents’ with ‘agent’, remove comma after dehydration.

Table 1. Reference 41. Replace ‘synthases’ with ‘synthase’, replace ‘hexose’ with ‘hexoses’

Table 1. Reference 50. Replace ‘prevent it from’ with ‘prevents’

Line 188. Rephrase. ‘to make potato adaptable’ sounds weird. Adaptation is a long-term evolutionary term. Replace ‘adaptable’ with ‘suitable’

Line 203. Replace ‘influenced’ with ‘affected’

Figure 1. What is the source of data? Reference? Are the temperatures listed Day or night temperatures? Tuber initiation and formation are very sensitive to night temperatures. Are the temperatures listed considered normal? The figure shows many ‘growth stages’. Several of the names listed in the x axis can not be considered growth stages (i.e., tuber yield’)

Line 257. Poor use of grammar ‘Potato that grown at places…

Author Response

Reviewer #2: 

Comment: The manuscript reviews the effect of heat and drought on tuber productivity, plant and tuber morphology, physiological processes, biochemical composition, and molecular genetics. It is a good review of the topics but does not provide much advice about germplasm screening and sources of heat and drought tolerance in cultivated potatoes.

Response: Authors are thankful for an encouraging and beneficial remarks and positive appreciation by the reviewer. The suggestions have been well taken and incorporated to the manuscript in track change mode. The reviewer has provided a valuable suggestion to emphasize germplasm screening and sources of stress tolerance. However, we focused and provided all the possible physiological, biochemical, and molecular responses of potato plant under high temperature and drought condition. In our future work we will surely incorporate the suggestions and prepare separate manuscript accordingly.

Comment: The title emphasizes ‘tropicalization’; however, tropicalization is not well defined. In addition, temperate climates also have issues with heat and drought (i.e., heat in Europe and US in recent years). Thus, ‘tropicalization of potato’ should be removed from the title, and ‘tropicalization’ should be revisited in the whole manuscript.

Response: The authors agree with the suggestion of the reviewer and we have removed the word “tropicalization” from the title. Moreover, we have revisited the tropicalization related phenomena across the manuscript.

Comment: Some original data or summaries should be included. For example, include maps illustrating regions in the world the authors are targeting (the review should expand beyond India; what about Africa, South America?). Summarize results from surveys about cultivated potato varieties that perform well in the targeted areas and add results from surveys from researchers in target areas about screening approaches to screen for heat and drought tolerance. Include a table showing selected areas where the highest yields are obtained and speculate why. Talk more deeply about yield losses due to heat and drought everywhere in the world.

Response: We agree with the reviewer’s concern and accordingly we have added the information on worldwide scenarios. The map has been included based on suggested lines. We have provided bibliometric analysis of the publication based on heat and drought stress in potato. Data for the bibliometric review on tropicalization of potato was collected from “Dimensions”. A bibliometric review is the need of the hour since it gives us idea about the intellectual framework and the collaborating authors and countries. Network visualization map was created using VoS viewer. The search engine used for the purpose of retrieval of data was “Heat Stress” AND “Potato”, “Abiotic Stress” AND “Potato”, “Heat and Drought Stress” AND “Potato”. The total number of data retrieved was 200 documents during the study period of 1980-2022. After data cleaning we were left with total number of 190 documents. The figure 1 shows where the research publications was highest for the topic on heat or drought stress of potato. It was noted that the countries we got in result were mostly the tropical countries as these are the areas where the prime focus was on tackling the challenge received due to high temperature and water deficit conditions.

Comment: Review the format used for references. Genus and species should be in italics; page numbers are missing in several references; inconsistent use of bold font…

Response: the format error and missing information in the bibliography has been corrected.

Comment: Request review of English language and grammar.

Response: The grammatical errors are carefully eliminated.

Comment: Below are some suggested edits, but the whole manuscript should be reviewed in depth.

Line 33: Replace ‘maintenance’ with ‘maintaining’

Response: The error is rectified.

Comment: Line 37: Replace ‘evaluation’

Response: The error is rectified.

Comment: Line 38 Replace ‘response’ with ‘responses’

Response: The word is modified.

Comment: Line 38-39: Replace ‘a prominent factor’ with ‘prominent factors’

Response: The error is rectified. (LN 39-40)

Comment: Line 41: Replace ‘these regions’ with ‘tropical regions’

Response: The word is modified

Comment: Line 43: All words included in the title are considered keywords. Thus, do not repeat them. Use different ones. For example, the scientific name of cultivated potato, heat tolerance, drought tolerance, climate change

Response: We are thankful for this suggestion; the key words are re-written.

Comment: Line 45: Move the introduction to the next line

Response: The line is shifted to the next page.

Comment: Line 46: If you refer to the cultivated tetraploid potato, you should say Solanum tuberosum spp. tuberosum L.

Response: The error is rectified.

Comment: Line 46: Replace ‘crop’ with ‘food crop’. Potato would rank four if corn is considered, Corn is mainly used for feed, thus if you indicate ‘food crop’ potato ranks third.

Response: We have mentioned it as ‘food crop’ in line 48.

Comment: Line 48: Indicate in parenthesis the year associated with the production year mentioned.

Response: We have mentioned the year as suggested.

Comment: Line 48: The increase will not be everywhere. In fact, in some areas it will decrease. Indicate in what areas production will increase (‘in several regions’)

Response: we acknowledge the suggestion and we have changed the sentence.

Comment: Line 49: Replace ‘well-known’ with ‘popular’

Response: The word is replaced.

Comment: Line 49: Replace ‘Indians’ with ‘many countries’

Response: The word is replaced.

Comment: Line 49: What about other countries like China, Russia, and the USA?

Response: The word is replaced.

Comment: Line 52: Replace ‘it’ with ‘potato’

Response: The word is replaced.

Comment: Line 53: Replace ‘this crop’ with ‘the potato crop’

Response: The word is replaced.

Comment: Line 56: Replace ‘the aforementioned issue is also the’ with ‘climate change will become a ’

Response: The sentence is modified as suggested.

Comment: Line 58: Domestication is the process of going from wild to domesticated. Thus, remove ‘cultivation’

Response: The error is rectified.

Comment: Line 58: Replace ‘was done’ with ‘took place’

Response: The word is replaced.

Comment: Line 58: Replace ‘hills’ with ‘highlands’

Response: The word is replaced.

Comment: Line 60: Remove ‘men’

Response: The word is replaced.

Comment: Lines 61 to 63. The use of ‘summer crop’ and ‘winter crop’ seems to be different than in the USA. In the USA, we use ‘fall crop’ if the crop is harvested I the fall and ‘summer crop’ if the crop is harvested in the fall. I believe you are using different terminology. To be consistent, indicate when the crop grows and is harvested. Refer to national USA statistics to see how fall and summer crops are reported. Most of the production in the USA is considered a fall Crop.

Response: The authors would like to thank the reviewer for suggestion. We have now rectified the sentence and modification has been made in line no 64-65.

Comment: Line 64: Replace ‘as a winter crop’ with ‘during the winter’

Response: The word is replaced.

Comment: Line 65: Review references for temperature for tuberization, temperatures below 20 are usually listed as required for tuberization. Temperatures above 20 at night reduce tuberization. You could end the phrase at below 22 C and start a new phrase saying that temperatures below 20 C are usually required for tuberization and list some references.

Response: We are thankful for this suggestion and the sentence has been added for better clarity.

Comment: Line 72: Remove ‘that’

Response: The word is removed.  

Comment: Line 74: Replace ‘recent past’ with ‘recent years’

Response: The word is removed. 

Comment: Line 74: Define ‘tropicalization’ in one phrase.

Response: We have defined ‘tropicalization’ in line 68-70.

Comment: Line 75: Expand to indicate where. ‘In tropical and sub-tropical countries?

Response: The suggestion has been added.

Comment: Line 75: Add ‘crop’ after ‘potato’

Response: The word is added.

Comment: Line 76: Replace ‘achieve better’ with ‘contribute to improve’

Response: The word is added.

Comment: Line 77: Replace ‘cues like’ with ‘factors such as

Response: The suggestion has been added.

Comment: Line 77: Add ‘flooding’?

Response: The word is added.

Comment: Line 78: Add comma before ‘and productivity’.

Response: The comma is added.

Comment: Line 82: Add comma before ‘and’

Response: The comma is added.

Comment: Line 92: Replace ‘a staple food’ with ‘staple foods’

Response: The word is added.

Comment: Line 95: Replace ‘response’ with ‘responses’

Response: The word is replaced.

Comment: Line 96: Replace ‘towards’ with ‘to’

Response: The word is replaced.

Comment: Line 96: Remove ‘and implication of knowledge’

Response: The word is removed.

Comment: Line 96: Replace ‘mechanism’ with ‘mechanisms’

Response: The word is replaced.

Comment: Line 97: Replace ‘response’ with ‘response to abiotic stresses’

Response: The word is replaced.

Comment: Line 98: Replace ‘by the introduction of tolerance’ with ‘with the goal of developing tolerant varieties

Response: The word is replaced.

Comment: Line 100: Replace ‘a detrimental effect’ with ‘detrimental effects’

Response: The word is replaced.

Line 103. Rephrase. The potato crop originated in borderline subtropical/alpine climate and performs best places with warm days and cool nights. Potato production in tropical and sub-tropical regions is challenging

Response: We are thankful for this correction. The word is rephrased for better clarity.

Comment: Line 104: Replace ‘the tropical region’ with ‘tropical regions’

Response: As suggested, the word is replaced.

Comment: Line 107: Remove ‘Presently,

Response: The word is removed.

Comment: Add references.

Response: The references are incorporated.

Comment: Lines 111-112. Remove ‘prevail in tropical areas’

Response: The sentence is removed.

Comment: Line 116: Remove ‘Unlike many crop species’

Response: The word is removed.

Comment: Line 122: Remove ‘The’

Response: The word is removed.

Comment: Line 123-124. Add period after ‘changes’

Response: The word is removed.

Comment: Line 124: Remove ‘in their’

Response: The word is removed.

Comment: Line 124: Start new phrase with ‘Growth…

Response: The word is added.

Comment: Line 124: Add ‘seriously affected’ after ‘development’

Response: The word is added.

Comment: Line 126 Replace ‘feature’ with ‘features’

Response: The word is replaced.

Comment: Line 131: Replace ‘summer’ with ‘the summer’

Response: The word is replaced.

Comment: Line 131: Remove ‘The’

Response: The word is replaced.

Comment: Line 132: Replace ‘imbalance’ with ‘imbalances

Response: The word is replaced.

Comment: Line 144: Replace ‘episode’ with ‘treatment’. Remove ‘treatment of’

Response: The word is replaced.

Comment: Line 147: Rewrite. Planting time affects potato yield.

Response: The sentence is rephrased.

Comment: Line 148: Replace ‘The high temperature’ with ‘High temperatures’ …

Response: The sentence is rephrased.

Comment: Line 149: Replace ‘is’ with ‘are’

Response: The word is replaced.

Comment: Line 149: Add ‘combination’ before ‘low’

Response: The word is added.

Comment: Line 150: Remove ‘The’

Response: The word is removed.

Comment: Line 153: Replace ‘the high’ with ‘high’

Response: The word is replaced.

Line 153: Replace ‘the potato’ with ‘potato’

Response: The word is replaced.

Line 158: Replace ‘the biomass’ with ‘biomass’

Response: The word is replaced.

Comment: Line 166: Remove ‘the’ in ‘the temperate’

Response: The word is removed.

Comment: Line 167: Remove ‘the’ in ‘the physiological’

Response: The word is removed.

Comment: Line 167: Do you mean ‘now adopted’ instead of ‘not adapted’?

Response: Yes, the error is rectified.

Comment: Line 168: Replace ‘the potato plant’ with ‘potato plants.

Response: The word is replaced.

Comment: Line 169: Replace ‘adapt’ with ‘acclimate’. Adapt is an evolutionary term.

Response: The word is replaced.

Comment: Line 170: Replace ‘on the sensitivity’ with ‘on the intensity’

Response: The word is replaced.

Comment: Line 170: Remove: of that is exposed’

Response: The word is removed.

Comment: Line 172: Explain the meaning of size (mass?)

Response: The word is modified as canopy mass.

Comment: Line 177: Explain the meaning of reproductive growth. Do you mean tuber enlargement?

Response: By means of reproductive growth we pointed out flower development.

Comment: Line 178: Replace ‘against’ with ‘to’

Response: The word is removed.

Comment: Line 180: Replace ‘The other’ with ‘Other’

Response: The word is removed.

Comment: Line 181: Mention in parenthesis the anatomical features you are referring to.

Response: The anatomical feature is added.

Comment: Table 1. Table heading. Remove ‘Different’

Response: The word is removed.

Comment: Table 1. Qualify biomass. Do you referent to above-ground plant biomass?)

Response: We have corrected its total biomass.

Comment: Table 1: Reduction of carotenoids refers to the plants? or the tubers?

Response: The correction is incorporated.

Comment: Table 1: Reference 21. Add ‘reduce’ before tuber filing. Replace ‘increase secondary tuber’ with ‘increase the production of secondary tubers’ Does russeting and cracking increase?

Response: The word is added.

Table 1. Reference 47. Remove the first comma.

Response: The comma is removed.

Comment: Table 1. Reference 48. Replace ‘which’ with ‘that’

Response: The word is removed.

Comment: Table 1. Reference 49. Increase in proline content? Replace ‘act’ with ‘acts’ and ‘agents’ with ‘agent’, remove comma after dehydration.

Response: The error is rectified.

Comment: Table 1. Reference 41. Replace ‘synthases’ with ‘synthase’, replace ‘hexose’ with ‘hexoses’

Response: The word is replaced.

Comment: Table 1. Reference 50. Replace ‘prevent it from’ with ‘prevents’

Response: The word is replaced.

Comment: Line 188. Rephrase. ‘to make potato adaptable’ sounds weird. Adaptation is a long-term evolutionary term. Replace ‘adaptable’ with ‘suitable’

Response: Thank you for the suggestion. The word is replaced.

Line 203. Replace ‘influenced’ with ‘affected’

Response: The word is replaced.

Figure 1. What is the source of data? Reference? Are the temperatures listed Day or night temperatures? Tuber initiation and formation are very sensitive to night temperatures. Are the temperatures listed considered normal? The figure shows many ‘growth stages’. Several of the names listed in the x axis cannot be considered growth stages (i.e., tuber yield’)

Response: As per the suggestion of the both the reviewers we have now removed the figure 1 from the text. We found the information provided was the repetition of the previously published data so we apologize the reviewer for this.

Dutt, S.; Manjul, A.S.; Raigond, P.; Singh, B.; Siddappa, S.; Bhardwaj, V.; Kawar, P.G.; Patil, V.U.; Kardile, H.B. Key players associated with tuberization in potato: potential candidates for genetic engineering. Crit. Rev. Biotechnol. 2017, 37, 942–957.

Comment: Line 257. Poor use of grammar ‘Potato that grown at places…

Response: The sentence is modified.

Round 2

Reviewer 2 Report

The manuscript has been significantly improved, but the construction of phrases and flow could be improved.

Title

Do not break the title. Replace ‘biochem-‘ with ‘biochemical’

Replace ‘of potato crop’ with ‘the potato crop’

The term ‘dynamics of’ in the tittle lacks meaning. Remove it.

Typically, the term ‘responses’ is followed by ‘of’ and later by ‘to’. Replace ‘under’ with ‘to’

Abstract

Start general and move into specifics. Mention that heat and drought stresses affect many potato-growing areas around the world. Point out where potatoes are planted in tropical areas (highlands?) and what are the main concerns, given global warming.

The phrase ‘This crop has the potential to provide a source of income besides the solution to food and nutritional security for developing countries of tropical regions’ is out of the context of heat and drought. The phrase could be removed or moved to another part where it makes sense.

Line 24. Replace ‘Potato is a tetraploid plant’ with ‘Most cultivated potatoes are tetraploid’.

Line 28. The indicated temperatures are optimum for tuber formation and enlargement, not plant growth. You could say ‘….optimum temperature for tuber production is 15-20 C)

The authors continue talking about tropical areas in the abstract. Heat and drought stress are not exclusive to tropical. Heat and drought affect many potato-growing areas and are of special concern in tropical areas. Adjust the abstract and introduction.

Later in the paper (lines 74-75) authors define tropicalization in the context of lower latitudes. Lower latitudes imply tuberization under short days or day-neutral, which is different from the northern latitudes' requirements (where heat and drought could also be problematic). The authors have not discussed photoperiod. Potato cultivars for tropical areas would have to combine the capacity of tuberizing under short-day or day-neutral conditions with tolerance to heat and drought. Indicate that in tropical areas, potatoes are typically grown in higher altitudes. However, with global warming, some areas are likely to suffer more frequent heat and drought spells. Provide more context (line 116). Also, incorporate the photoperiod aspect in other sections, including the conclusion.

Remove figure 1 and associated text (lines 181 to 192). Replace with a map showing potato growing areas exposed to heat and drought stress. The authors say (lines 190-192) that ‘…the countries we got in results were mostly the tropical countries….’ The map misses many tropical countries where potatoes are grown, except India and South Africa (subtropics).

Line 70: remove ‘fall crop’

The conclusion is weak in providing recommendations for tropicalization. It seems like the main recommendation is the selection of early maturing germplasm (in what breeding programs? Where? Why?). There is no discussion of screening approaches/methodologies. Also, not much is mentioned in the text in terms of sources of heat and drought-tolerant germplasm.

References

Lines 811: italics in the scientific name (review all references: lines 811, 821, 890….), add space before L

Line 1003: Use lowercase

Several references are in the title case (the first letter of each word in upper case). Keep in uppercase only the first word. (review all references: lines 806, 894, 914, 927, 933, 1067, 1079, 1082…)

In a few places, it says Science (80-). Review and fix line 1090, line1098

Line 923: do not use ‘et al’. List all authors.

Author Response

Reviewer #2

Comment: The manuscript has been significantly improved, but the construction of phrases and flow could be improved.

Response: The reviewer would like to sincerely acknowledge the reviewer for positive and constructive comments towards the improvement of the manuscript. As per the suggestion we have improved the manuscript.

Comment: Do not break the title. Replace ‘biochem-‘ with ‘biochemical’

Replace ‘of potato crop’ with ‘the potato crop’

The term ‘dynamics of’ in the tittle lacks meaning. Remove it.

Typically, the term ‘responses’ is followed by ‘of’ and later by ‘to’. Replace ‘under’ with ‘to’

Response: As per the suggestion, we have modified the title of the manuscript.

Comment: Abstract

Start general and move into specifics. Mention that heat and drought stresses affect many potato-growing areas around the world. Point out where potatoes are planted in tropical areas (highlands?) and what are the main concerns, given global warming.

Response: As per the suggestion, the abstract section is now modified accordingly.

Comment: The phrase ‘This crop has the potential to provide a source of income besides the solution to food and nutritional security for developing countries of tropical regions’ is out of the context of heat and drought. The phrase could be removed or moved to another part where it makes sense.

Response: We have eliminated this line.

Comment: Line 24. Replace ‘Potato is a tetraploid plant’ with ‘Most cultivated potatoes are tetraploid’.

Response: We have replaced this line with the suggested phrase.

Comment: Line 28. The indicated temperatures are optimum for tuber formation and enlargement, not plant growth. You could say ‘….optimum temperature for tuber production is 15-20 C)

Response: As per the suggestion, the abstract section is now modified accordingly.

Comment: The authors continue talking about tropical areas in the abstract. Heat and drought stress are not exclusive to tropical. Heat and drought affect many potato-growing areas and are of special concern in tropical areas. Adjust the abstract and introduction.

Response: As per the suggestion we have signifincantly improved abstract and introduction section.

Comment: Later in the paper (lines 74-75) authors define tropicalization in the context of lower latitudes. Lower latitudes imply tuberization under short days or day-neutral, which is different from the northern latitudes' requirements (where heat and drought could also be problematic). The authors have not discussed photoperiod. Potato cultivars for tropical areas would have to combine the capacity of tuberizing under short-day or day-neutral conditions with tolerance to heat and drought. Indicate that in tropical areas, potatoes are typically grown in higher altitudes. However, with global warming, some areas are likely to suffer more frequent heat and drought spells. Provide more context (line 116). Also, incorporate the photoperiod aspect in other sections, including the conclusion.

Response: As per the suggestion of the esteemed reviewer we have improved the text. Please see section 3.3 LN 264-277. Further the physiological and molecular aspect is also included in LN 303-322.

Comment: Remove figure 1 and associated text (lines 181 to 192). Replace with a map showing potato growing areas exposed to heat and drought stress. The authors say (lines 190-192) that ‘…the countries we got in results were mostly the tropical countries….’ The map misses many tropical countries where potatoes are grown, except India and South Africa (subtropics).

Response: As per the suggestion Figure 1 has been replaced with an updated map showing potato growing area exposed to heat and drought stress.

Comment: Line 70: remove ‘fall crop’

Response: We have removed the given word.

Comment: The conclusion is weak in providing recommendations for tropicalization. It seems like the main recommendation is the selection of early maturing germplasm (in what breeding programs? Where? Why?). There is no discussion of screening approaches/methodologies. Also, not much is mentioned in the text in terms of sources of heat and drought-tolerant germplasm.

Response: The authors acknowledge the reviewer for suggestions. We have improved the conclusion section.

Comment: References

Lines 811: italics in the scientific name (review all references: lines 811, 821, 890….), add space before L

Response: As per the suggestion all scinetific names are now italicized.

Comment: Line 1003: Use lowercase

Response: The suggestion has been incorporated.

Comment: Several references are in the title case (the first letter of each word in upper case). Keep in uppercase only the first word. (review all references: lines 806, 894, 914, 927, 933, 1067, 1079, 1082…)

Response: As per the suggestion the reference has been revisited and modified accordingly

Comment: In a few places, it says Science (80-). Review and fix line 1090, line1098

Response: The modification has been made in reference no. 19.

Comment: Line 923: do not use ‘et al’. List all authors.

Response: We have arranged the reference according to the MDPI Plants format.
